# Molecular and Functional Key Features and Oncogenic Drivers in Thymic Carcinomas

**DOI:** 10.3390/cancers16010166

**Published:** 2023-12-29

**Authors:** Serena Barachini, Eleonora Pardini, Irene Sofia Burzi, Gisella Sardo Infirri, Marina Montali, Iacopo Petrini

**Affiliations:** 1Department of Translational Research and of New Surgical and Medical Technologies, University of Pisa, 56126 Pisa, Italy; 2Department of Clinical and Experimental Medicine, University of Pisa, 56126 Pisa, Italy

**Keywords:** thymic epithelial tumors, thymic carcinoma, GTF2I, hallmarks of cancers, angiogenesis, immunotherapy, and apoptosis

## Abstract

**Simple Summary:**

Thymic carcinomas are the most aggressive thymic epithelial tumors. While thymic carcinomas do not have a single specific cause, various aspects of their growth have been evaluated at the molecular level. In this review, we organize the recent findings according to what have been defined as the hallmarks of cancer. The identification of the molecular alterations that drive the growth of these tumors is crucial for the development of effective target therapies. Inactivation of CDKN2A, TP53, and CDKN2B by somatic mutation or deletion is most common in thymic carcinomas. In advanced tumors, mutations in genes that regulate epigenetic and chromatin remodeling can occur. On the contrary, mutations in tyrosine kinase receptors and other oncogenes are only occasional, with those of KIT being the most common and present in only 10% of thymic carcinomas. Thymic carcinomas present an enhanced and aberrant vasculature that has been targeted in clinical trials with promising results. Additionally, thymic carcinomas evade the control of the immune system, with some tumors showing a response to immune checkpoint inhibitors.

**Abstract:**

Thymic epithelial tumors, comprising thymic carcinomas and thymomas, are rare neoplasms. They differ in histology, prognosis, and association with autoimmune diseases such as myasthenia gravis. Thymomas, but not thymic carcinomas, often harbor GTF2I mutations. Mutations of CDKN2A, TP53, and CDKN2B are the most common thymic carcinomas. The acquisition of mutations in genes that control chromatin modifications and epigenetic regulation occurs in the advanced stages of thymic carcinomas. Anti-angiogenic drugs and immune checkpoint inhibitors targeting the PD-1/PD-L1 axis have shown promising results for the treatment of unresectable tumors. Since thymic carcinomas are frankly aggressive tumors, this report presents insights into their oncogenic drivers, categorized under the established hallmarks of cancer.

## 1. Introduction

Thymic epithelial tumors are rare, with an incidence of 0.32 cases per 100,000 people per year [1]. Thymic epithelial tumors encompass thymomas, thymic carcinomas, and thymic neuroendocrine neoplasms. Thymic carcinomas account for 14–22% of all thymic epithelial tumors and are highly aggressive tumors with poorer survival rates compared to thymomas [2]. The median age at presentation is 54–65.5 years, but thymic carcinomas have been diagnosed in patients ranging in age from 12 to 96 years old. Thymic carcinomas are prevalent in males [3]. Thymic carcinomas are frequently diagnosed at advanced stages, making surgical resection unfeasible. They tend to grow locally, invading mediastinal structures and spreading into the pleural cavity. In contrast to thymomas, thymic carcinomas rapidly progress through lymph nodes or hematogenous metastases [2]. Notably, thymic carcinomas exhibit a higher Ki67 index than thymomas [4]. Myasthenia gravis is exceptionally rare in patients with thymic carcinomas [2].

The histological appearance of thymomas resembles that of normal thymic structures, while thymic carcinomas exhibit features akin to carcinomas originating in other organs. Therefore, thymic carcinomas are further classified into 14 subgroups based on their histological features and molecular aberrations (see Table 1) [2]. Most commonly, thymic carcinomas are squamous cell carcinomas (70–80%). Immunohistochemical staining for CD5 and CD117 (KIT) is helpful in distinguishing them from squamous carcinomas originating in other organs, such as the lungs. The expression of cancer testis antigens and melanoma antigens is more frequent in thymic carcinomas than in thymomas. GAD1 thesis antigen overexpression is a poor prognostic factor in thymic carcinoma [3]. Other types of thymic carcinomas are rare and can include histological aspects of basaloid tumors, lymphoepithelioma-like carcinomas, or adenocarcinomas. Lymphoepithelioma-like carcinoma is associated with EBV infection similarly to what is observed in nasopharynx tumors [5,6]. NUT-BRD4 fusion genes are observed in a subset of highly aggressive tumors originating from organs located along the body’s midline. While the thymus is situated across the body’s midline, only a few cases of poorly differentiated thymic carcinoma with NUT-BRD4 fusion genes have been described in the literature. NUT rearrangements most frequently occur with BRD4, but additional partners have been described, including BRD3 and NSD3 [7]. Other gene fusions characterize specific histotypes of thymic carcinomas, such as EWSR1-ATF1 in salivary gland-type hyalinizing clear cell carcinoma or CRTC1-MAML2 in some mucoepidermoid carcinomas of the thymus [2]. Thymic carcinomas and thymomas are considered distinct diseases due to differences in their clinical behavior, histological features, and molecular aberrations. Thymic carcinomas exhibit a higher mutation rate compared to thymomas [8]. Indeed, the tumor mutational burden (TMB) is significantly higher in thymic carcinomas compared to thymomas [9,10]. GTF2I mutations are common in thymomas, especially in A and AB histotypes, and uncommon in thymic carcinomas (8–25%), where TP53 mutations are prevalent [11,12]. Reviewing the literature on somatic mutations in thymic epithelial tumors, we constructed a concordance table and observed that GTF2I mutations co-occurred with HRAS, TTN, and UNC93B1 mutations. Conversely, GTF2I mutations and TP53 mutations tend to be mutually exclusive, characterizing A and AB thymomas and B3 thymomas and thymic carcinomas, respectively [13]. Mutations of BRCA2, SETD2, PBRM1, and CDKN2A significantly co-occurred with TP53, while a trend for the absence of co-occurrence with GTF2I was also observed for CDKN2A, BAP1, CYLD, and KIT mutations. The analysis of mutational signatures in GTF2I and TP53 mutant tumors showed similarities in the spontaneous deamination of the 5-methylcytosine signature but discrepancies in the enrichment of signature 5 (an unknown function) and the mismatch repair signature [13].

There are tumors where areas of both thymomas and thymic carcinomas coexist, leading some researchers to argue for a continuum between the two tumor types. However, this view is not universally accepted. The different types of somatic mutations suggest that thymic carcinomas and thymomas are distinct. Data from other tumors suggests that a common trunk of mutations occurs in the earliest steps of carcinogenesis; these mutations are shared by all cancer cells in a single tumor. Mutations that develop later are present in subclones, contributing to the heterogeneity typically achieved in more advanced tumors [14]. Data on the heterogeneity of somatic mutations in thymic epithelial tumors are limited. Recently, it has been demonstrated that medullary and cortical thymic epithelial cells originate from a common precursor and that adult thymic epithelial cells exhibit impaired differentiation [15]. Upon neoplastic transformation, thymic epithelial cells lose their differentiative potential and can reorient their polarization to form cells with cortical, medullary, or mixed features. Hybrid gene expression profiles have been observed in experimental transgenic models of thymic epithelial tumors [16,17], suggesting that mixed histological features could be dependent on differentiative dysregulation. A significant portion of squamous thymic carcinomas express markers associated with tuft cells, such as POU2F3, L1CAM, and GFI1B [18,19]. Thymic tuft cells are a subset of medullary epithelial cells that impact innate immunity [20,21]. The expression of tuft cell markers in some thymic carcinomas suggests that they may originate from medullary tuft cells. The presence of KIT expression in normal medullary tuft cells and in thymic carcinomas appears to support this hypothesis. Gene expression analysis of thymic epithelial tumors reveals that carcinomas with tuft cell-like features tend to cluster together and differ from other thymic carcinomas that lack this phenotype [18,19].

Since thymic carcinomas are rare diseases, it is economically unappealing for companies to invest significantly in clinical and biological research for the development of specific drugs. Consequently, scientists have pursued two primary lines of research. First, genome-wide screenings of somatic mutations have been conducted in thymic epithelial tumors. Second, oncogenic drivers that can be targeted with therapies used in other types of cancer have been assessed. Tyrosine kinase inhibitors and immunotherapy have demonstrated clinical activity in thymic carcinomas but not in thymoma. For instance, in phase 2 clinical trials involving patients with previously treated thymic carcinomas, sunitinib and lenvatinib have achieved objective response rates of 22–26% and 38%, respectively [22,23,24]. Sunitinib and lenvatinib are tyrosine kinase inhibitors targeting PDGFR, VEGFR, KIT (sunitinib), and VEGFR, FGFR, RET, and KIT (lenvatinib). Recently, a phase 2 clinical trial of ramucirumab combined with paclitaxel and carboplatin in treatment-naïve metastatic thymic carcinomas reported a promising objective response rate of 57.6%, surpassing standard chemotherapy [25]. The inclusion of ramucirumab, an anti-VEGFR2 antibody, underscores the significance of angiogenesis in the growth of thymic epithelial tumors. Immunotherapy has shown effectiveness in thymic carcinomas but is fraught with danger in thymomas due to the high incidence of severe immune-related toxicities. Pembrolizumab has achieved an objective response rate of 23% in previously treated thymic carcinomas, with some durable responses. The efficacy of immunotherapy has been further validated with other anti-PD-1 and PD-L1 antibodies, indicating the importance of evading immune surveillance in thymic carcinomas [26].

Given the rarity of the disease and the heterogeneous and non-systematic scientific production in recent years, we aim to examine the recent literature on molecular alterations in thymic carcinomas. This review will be organized according to the well-established hallmarks of cancer in order to understand which of these have been well-characterized and which still represent perspectives for innovative research.

## 2. Materials and Methods

The literature review were conducted using the PubMed database. The search was performed with the keywords ‘thymic carcinoma’ and ‘thymic epithelial tumors’. Articles published between 2023 and 2000 were included if they reported molecular aberrations in thymic carcinomas. We specifically focused on references related to the hallmarks of cancer, as described in 2011. Epigenetic abnormalities, including micro-RNA, long noncoding RNA, and methylation, were excluded unless directly associated with a specific hallmark of cancer. Additionally, data included in the fifth edition of the World Health Classification of Tumors: Thoracic Tumors was incorporated. The data predominantly pertain to squamous cell carcinomas of the thymus; specific aberrations of other types of thymic carcinomas are poorly understood and are excluded from this review.

## 3. Oncogenic Drivers in Thymic Carcinomas

In thymic carcinomas, there is no known genetic aberration that can be responsible for the oncogenic transformation and therefore is not an elective target for treatment. On the contrary, in thymomas, GTF2I is an oncogene with recurrent mutations that are present in most of the A/AB thymomas and in a relevant number of B histotypes. Over the last decades, several attempts have been made to understand the molecular aberrations of thymic carcinomas, both with sequencing screening and histological analysis. Limited data are available regarding the functional characterization of the mutations identified in thymic carcinomas, mainly because of the paucity of cell lines available. Some insights can be obtained from patients who have been treated with targeted therapies within clinical trials, providing an in vivo functional characterization of these tumors.

Genomic characterization of 414 thymic carcinomas using the Foundation Medicine platform revealed that the most frequently altered genes were CDKN2A (39.9%), TP53 (30.2%), CDKN2B (24.6%), BAP1 (8.2%), TET2 (8.0%), KIT (8.0%), SETD2 (7.7%), NFKBIA (7.7%), ASXL1 (7.0%), and KMT2D (6.0%). Tumors with high TMB (≥10 mutations/Mb) and microsatellite instability (MSI) were observed in 7.0% and 2.3% of cases, respectively [27]. The results appear consistent with previous reports. In a series of 48 thymic carcinomas assessed for mutations in 50 cancer-related genes, the TP53 pathway was the most frequently affected (20.4%), followed by the receptor tyrosine kinase/RAS pathway (18.5%), and the PI3K pathway (5.6%) [28].

Since thymic carcinomas are highly malignant tumors, it is reasonable to evaluate progress according to each hallmark of cancer.

## 4. Hallmarks of Cancer in Thymic Carcinomas

### 4.1. Sustaining Proliferative Signaling

Whole exome sequencing data from 9125 tumors profiled by The Cancer Genome Atlas (TCGA) indicate that the most commonly disrupted pathways in cancer are the receptor tyrosine kinase/Ras pathway, cell cycle, and PI3K pathways [29]. Mutations in genes associated with these pathways provide the majority of the targets used for biological treatments in clinical practice.

KIT is expressed in 80–86% of thymic carcinomas [30,31]. Mutations of KIT have been described in 10% of thymic carcinomas, and these mutations are detailed in Table 2 [32]. While drugs like imatinib, sunitinib, and other tyrosine kinase inhibitors with inhibitory activity against KIT have shown anecdotal responses in a few patients, phase II trials of imatinib in non-selected thymic carcinomas have yielded disappointing results [33,34]. EGFR expression is lower in thymic carcinomas compared to thymomas [35], and somatic mutations in EGFR are infrequent, which explains the limited efficacy of EGFR inhibitors in thymic epithelial tumors [36,37]. Occasional mutations in FGFR2 and FGFR3 have been reported [38], as well as mutations in PDGFRA [28].

IGF-1 is a trophic factor for the normal thymus, and the decline in thymic function that occurs with age corresponds to the decrease in IGF-1 levels in the bloodstream. Thymic epithelial cells express the IGF-1 receptor (IGF-1R), and the addition of IGF-1 to human thymic epithelial cell lines enhances the interaction between thymocytes and epithelial cells by modulating the expression of cell adhesion molecules and promoting the production of extracellular matrix proteins [51]. Moreover, IGF-1 has been demonstrated to induce proliferation in cultured human thymic epithelial cell lines. IGF-1R is highly expressed in thymic carcinomas (92%) [52], and occasional mutations of IGF1R, including G596V, have been reported [53]. In a patient with acromegaly and a thymoma, the surgical removal of the GH-producing tumor resulted in a reduction in the size of the thymoma along with a decrease in plasma IGF-1 levels [54]. Unfortunately, a phase II trial of cixutumumab, an anti-IGF-1R antibody, yielded modest results, with only 14% of objective responses [55].

The relevance of the interaction between the thymic stroma and extracellular matrix is underscored by the overexpression of focal adhesion kinase (FAK) in thymic epithelial tumors, especially in highly aggressive subtypes like type B2 and B3 thymomas and thymic carcinomas [56].

The importance of signaling pathways downstream of tyrosine kinase receptors is evident from the detection of HRAS, NRAS, and KRAS mutations. In thymic carcinomas, RAS mutations are more commonly missense, affecting the Q61 codon. Occasional mutations in genes related to the PI3K complex have been observed in thymic carcinomas, and the effectiveness of PI3K inhibitors has been confirmed in thymic carcinoma cell lines [57]. PTEN protein is not detected in the normal thymus by immunohistochemistry but is expressed in type A thymoma and carcinoma cells. No PTEN mutations or promoter methylation have been reported in thymic carcinomas [58]. However, AKT phosphorylation is increased in aggressive thymic epithelial tumors, especially in thymic carcinoma, and is associated with a poor prognosis [59].

Thymic carcinomas and B3 thymomas exhibit increased expression of WNT4 compared to the normal thymus [60]. The absence of WNT4 suppresses fetal and postnatal thymic expansion, leading to a decrease in thymic epithelial cell numbers, an alteration of the medullary-to-cortical thymic epithelial cell ratio, and a loss of immature thymocyte precursors [61]. In short-term 2D culture, neoplastic cells exhibited a decrease in WNT4 expression and secretion. However, this decline was not observed in 3D spheroids. Under these conditions, or with recombinant WNT4 supplementation, the growth of thymic epithelial cells was accompanied by an increased expression of RAC1 and JNK. The downregulation of WNT4 by shRNA induced cell death in primary cultures of B3 thymomas and resulted in decreased RAC1 expression without JNK phosphorylation. Moreover, RAC1 and JNK phosphorylation were diminished by the pharmacological inhibition of NFκB in primary cultures of neoplastic thymic epithelial cells [60].

The transcription factor SOX2 maintains self-renewal and pluripotency in undifferentiated embryonic stem cells. Interestingly, the mRNA expression of SOX2 in fresh tumor tissues from thymic carcinomas was significantly higher than that observed in thymomas. Higher expressions of SOX2 and IGF-1 proteins were markedly elevated in thymic carcinomas compared to thymomas, as observed through immunohistochemistry. Therefore, SOX2 and IGF-1 expression are higher in more aggressive TETs, indicating a poorer prognosis [62].

### 4.2. Evading Tumor Suppression

According to the Foundation Medicine database, the most frequently altered tumor suppressor genes in thymic carcinoma are CDKN2A (39.9%), TP53 (30.2%), and CDKN2B (24.6%) [27], combining deletions and mutations. TP53 is the most commonly mutated gene in cancer. The relatively high incidence of TP53 mutations in thymic carcinomas has been consistently reported in various studies [10,12]. The presence of mutant TP53 has been linked to poorer overall survival in thymic carcinomas [8,46,63]. In the normal human thymus, TP53 expression is observed in scattered thymic epithelial cells, primarily in the subcapsular cortical region, while p63 and p73 are expressed by the vast majority of cortical and medullary cells. Thymocytes do not express the p53, p63, or p73 proteins [64]. Thymic carcinomas exhibit more intense p63 staining compared to thymomas [65], and all thymic epithelial tumors demonstrate higher expression of p73 compared to normal thymus tissue [66].

CDKN2A and CDKN2B loci are closely mapped within the short arm of chromosome 9 (9p21.3). CDKN2A codifies for two relevant tumor suppressor genes: p16INK4, which inhibits cell cycle progression in G1, and p14ARF, which controls the expression of MDM2 and therefore TP53. CDKN2B encodes p15INK4B, an inhibitor of the cell cycle. p15INK4B blocks cells in the G1 phase by binding to CDKN4 and CDK6 and inhibiting their activity. Therefore, this locus on chromosome 9 is a point of weakness in the human genome in the case of deletion because it contains several tumor suppressor gene loci close together. Indeed, focal deletions of CDKN2A and CDKN2B are commonly observed in aggressive tumors originating from various organs [67]. We observed focal deletions of the CDKN2A and CDKN2B loci in thymic carcinomas and B3 thymomas [68]. Tumors with a focal deletion of 9p21.3 do not express p16INK4. The presence of this focal deletion and the absence of p16INK4 expression are poor prognostic factors in thymic epithelial tumors [68,69]. Moreover, inactivating mutations of CDKN2A have been described in thymic carcinomas [10,12,27], and p16INK4 expression can be suppressed by methylation of its promoter in thymic epithelial tumors [68].

Cyclin-dependent kinase (CDK) inhibitors p16, p21, p27, and p57 play differential roles in the proliferation, survival, and differentiation of thymic epithelial cells and thymocytes. Immunostaining of p21 has been detected only in a small proportion of epithelial cells within thymomas [4]. On the contrary, thymic carcinomas express more p21 than B3 thymomas [70].

The number of cells expressing p16 increases with age, consistent with the understanding that heightened p16 expression is associated with cellular senescence [71]. β-galactosidase activity and the evaluation of DNA damage using c-H2AX immunostaining demonstrate an increase of senescent thymic epithelial cells in older mice [72,73].

RB1 expression and RB1 phosphorylation have been reported in most thymic epithelial tumors, including thymic carcinomas [74]. Inactivating mutations and deletions of RB1 have been described in thymic epithelial tumors but appear relatively uncommon. However, the inactivation of the RB1 pathway could be a relevant event in the carcinogenesis of thymic tumors. Indeed, transgenic mice expressing SV40 large T antigen or overexpressing E2F in thymic epithelial cells develop thymomas that progress into thymic carcinomas [68,75].

### 4.3. Activating Invasion and Metastasis

Thymic carcinomas often metastasize to lymph nodes and other organs. The process of metastatic spread is recognized as a multistep progression in which tumor cells undergo epithelial-to-mesenchymal transition (EMT), a critical phenomenon that enables their infiltration and movement into the surrounding tissues. To reach the bloodstream and embark on their journey to distant organs through intravasation, tumor cells must produce metalloproteinases, enzymes capable of breaking down the extracellular matrix of the surrounding tissues [76].

Immunohistochemistry has revealed reduced expression of E-Cadherin and increased expression of N-Cadherin, TWIST, and SNAIL in thymic carcinomas compared to thymomas, suggesting that thymic carcinomas are more likely to undergo EMT with an increased metastatic potential [77]. Thymic carcinomas displaying features indicative of an EMT status, such as reduced E-Cadherin expression and increased N-Cadherin, TGF-B, and vimentin expression, tend to have poorer disease-free survival after resection [78]. Notably, induction chemotherapy appears to increase the number of cells expressing EMT markers and PD-L1 [78]. Fibronectin is a marker of EMT, and the ED-B isoform is expressed in the stroma of metastatic thymic epithelial tumors [79]. Intriguingly, knocking down LINC00174 in the T1889 thymic carcinoma cell line has been shown to reduce N-Cadherin and increase E-Cadherin levels compared to control cells. LINC00174 is a long noncoding RNA that can sequester miR-145-5p, a microRNA involved in the regulation of EMT [80]. MiR-145-5p is epigenetically downregulated in thymic tumors, contributing to increased motility of T1889 cells in vitro and alterations in their lipid metabolism [81].

Matrix metalloproteinases (MMPs) play a role in thymic tumors. MMP-2 and MMP-7 are predominantly expressed in type B3 thymoma and thymic carcinoma, while MMP-9 is preferentially expressed in B2 thymomas. Moreover, MMP-2 and MMP-7 are more expressed in more advanced-stage tumors. The gelatinolytic activity of MMP-2 increases with invasiveness in thymic epithelial tumors, and the expression of MMP-2 is indicative of a poor outcome [82,83,84].

### 4.4. Enabling Replicative Immortality

The proliferative capacity of cells is limited by the onset of senescence and telomere shortening. Senescence is triggered by factors such as oxidative stress, DNA damage, enhanced oncogenic signaling, and mitochondrial dysfunction. These factors result in the expression of cell cycle inhibitors like p16INK4, p21, p27, and p57 proteins [85]. We already discussed how thymic carcinomas downregulate these proteins to overcome the growth arrest due to senescence. To prevent telomere shortening and subsequent chromosome fusions, cancer cells upregulate the expression of hTERT in 85% of tumors. The remaining 15% rely on alternative lengthening of telomeres [86]. Limited data are available on the mechanisms employed by thymic carcinomas to achieve replicative immortality. Telomerase activity has been detected in all thymic epithelial tumors, but it tends to be higher in thymomas due to the abundance of thymocytes. In patients with thymic carcinoma, telomerase activity positively correlates with tumor stage [87]. No mutations of the hTERT promoter were observed in 14 thymic carcinomas [88].

### 4.5. Inducing Angiogenesis

The vasculature in thymomas and thymic carcinomas exhibits differences. A and AB thymomas display a dense vasculature characterized by small, capillary-like vessels with smaller diameters. In contrast, type B1, B2, and B3 thymomas exhibit progressively larger vessels and a less dense vascular network. Thymic carcinomas, on the other hand, present the least dense vascular network [89]. Raica and colleagues categorized tumor blood vessels in thymomas as immature, intermediate, and mature, and they observed a correlation between blood vessel type and endothelial cell proliferation with invasiveness [90]. Additionally, Tomita et al. found a link between tumor angiogenesis and the invasiveness of thymomas [91].

A comparison of protein expression between 84 low-risk (A, AB, and B1) and 116 high-risk thymic epithelial tumors (B2, B3, and thymic carcinomas) using tissue microarray analysis revealed an increased expression of VEGFA, VEGFC, and VEGFD, as well as the receptors VEGFR1, VEGFR2, and VEGFR3 in high-risk thymic epithelial tumors [92]. In an independent analysis, VEGFR1 and VEGFR2 expression were detected in vascular endothelial cells across all analyzed tumor subtypes. VEGFR1 was expressed in tumor epithelial cells, with higher levels in A, B3, and thymic carcinomas. In contrast, VEGFR2 expression was more pronounced in the epithelial cells of B3 thymomas and thymic carcinomas [89]. Elevated serum levels of VEGFA have been observed in thymic epithelial tumor patients, and there is a reported association between VEGF expression and the invasiveness of thymic epithelial tumors [93]. The stronger efficacy of sunitinib and lenvatinib in thymic carcinomas compared to thymomas underscores the importance of tumor angiogenesis in these tumors [22,23,24]. Promising results with ramucirumab, an anti-VEGFR2 agent, further support the significance of this receptor in thymic carcinoma growth and angiogenesis [25].

### 4.6. Resisting Cell Death

The expression of BCL2 in thymic carcinomas is different from that in thymomas. Epithelial cells in thymic carcinomas uniformly express BCL2, while B thymomas typically do not express BCL2. In contrast, A histotype thymomas can exhibit varying levels of BCL2 positivity, ranging from 60 to 100% in different reports [4,94,95,96]. Amplification of the BCL2 locus has been observed in thymic carcinomas using array comparative genomic hybridization (CGH). Tumors with BCL2 amplification had an overexpression of the protein, as demonstrated by Western blot analysis [68]. Survival of thymic epithelial tumor cell lines (T1889, T1682, and TY82) depends on BCL2 and MCL1 expression, as demonstrated by siRNA knockdown [68]. MCL1 is essential for the maintenance of cortical and medullary thymic epithelial cells, suggesting its potential relevance in thymic epithelial tumor development [68,97]. Commonly, both MCL1 and BCL2 are simultaneously expressed in thymic carcinomas [68]. In thymic epithelial tumor cell lines and xenografts, Gx15-070, a pan-BCL2 family protein inhibitor, effectively suppresses cell growth by inducing autophagy-dependent necroptosis [68]. Indeed, expression of other anti-apoptotic BCL2-family molecules has been documented in thymic epithelial tumors. BCL-XL is frequently expressed in thymomas (50%) and in thymic carcinomas (90%) [4,98,99]. In T1889 cells, the inhibition of MCL1 alone is insufficient to trigger apoptosis. However, when the inhibition of both MCL1 and BCL-XL is combined, it leads to caspase-dependent cell death [98].

In a gene expression analysis by microarray, thymic carcinomas were found to upregulate two anti-apoptotic genes (BIRC3 and SCYA20), two pro-apoptotic genes (PMAIP1 and MYC), and downregulate the pro-apoptotic gene MTCH2 when compared to B3 thymomas. T1889 thymic carcinoma cell lines express BIRC-3, and its knock-down by siRNA induces apoptosis [100].

In the normal thymus, FAS expression is primarily observed in epithelial cells, with weak FAS-ligand expression in the medullary epithelium and Hassal’s corpuscles. In thymic epithelial tumors, FAS was expressed in epithelial cells and thymocytes, whereas FASL was not expressed in thymocytes. The expression of FASL in epithelial cells varies among different thymic epithelial tumor subtypes [101].

When the extrinsic apoptotic pathway is activated, it leads to the formation of DISC, a multi-protein complex consisting of members from the death receptor family, which induces apoptosis by activating caspase 3. C-FLIP binds to the DISC complex and can inhibit apoptosis, autophagy, and necroptosis, promoting cell survival [102]. C-FLIP is overexpressed in over 90% of thymomas and thymic carcinomas when compared to normal thymic tissue. Additionally, while C-FLIP expression decreases in the normal thymus during aging, this decline is not observed in thymomas [103].

### 4.7. Deregulating Cellular Energetics

Using high-resolution magic-angle spinning 1H nuclear magnetic resonance (HRMAS 1H-NMR) spectroscopy, 37 metabolites were identified in 15 thymic epithelial tumors, including 3 thymic carcinomas. Aggressive thymic epithelial tumors (B2, B3, and thymic carcinomas) exhibited more metabolites and demonstrated increased activation of six metabolic pathways. These pathways had functional implications related to trans-sulfuration, homocysteine and tricarboxylic acid cycles, the management of reactive oxygen species (ROS), and glycolysis. Analysis of TCGA data showed differential gene expression in aggressive thymic epithelial tumors compared to indolent ones (A, AB, B1), with an enrichment of pathways associated with lactate (glycolysis), alanine (the TCA cycle and the alanine/aspartate/glutamate pathway), and glutathione, as revealed by Gene Set Enrichment Analysis (GSEA). Aggressive thymomas and thymic carcinomas displayed elevated lactic acid levels, indicative of the Warburg effect, a metabolic shift in energy production that involves cancer cells favoring glycolysis even in the presence of oxygen [104].

The glucose transporter GLUT1 is more highly expressed in thymic carcinomas than in thymomas or the normal thymus. GLUT1 expression correlates with fluorine-18 deoxyglucose uptake, as observed in positron emission tomography. Consequently, fluorine-18 deoxyglucose uptake is typically observed in thymic carcinomas and in most B2 and B3 thymomas but rarely in A, AB, and B1 thymomas [105,106].

L-type amino acid transporter 1 (LAT1, SLC7A5) plays a crucial role in incorporating essential amino acids into cells and is prominently utilized by various human cancers. LAT1 was found to be strongly expressed in human thymic carcinoma, and its inhibition significantly suppressed leucine uptake and the growth of Ty82 cells. This suggests that thymic carcinoma takes advantage of LAT1 as a quality transporter [107].

The upregulation of LINC00174 expression in thymic tumor tissue sequesters miR-145-5p, allowing the expression of its target genes UBAC1, SYBU, FEM1B, and SCD5. A bioinformatic analysis indicated that these genes are enriched in lipid metabolism. Knocking down LINC00174 or SCD5 in the T1889 cell line resulted in a reduction of lipid droplet content [81].

### 4.8. Genome Instability and Mutations

Thymic carcinomas exhibit a higher mutation rate than thymomas [8]. The incidence of high (≥10 mutations/Mb) and high MSI in thymic carcinomas was 7.0% and 2.3%, respectively. The most altered genes in 414 thymic carcinomas evaluated using the Foundation Medicine assay were CDKN2A (39.9%), TP53 (30.2%), CDKN2B (24.6%), BAP1 (8.2%), TET2 (8.0%), KIT (8.0%), SETD2 (7.7%), NFKBIA (7.7%), ASXL1 (7.0%) and KMT2D (6.0%) [27].

In 2018, we reviewed the literature and collected somatic mutations of thymic epithelial tumors reported in analyses performed using next-generation sequencing [13]. Among 339 thymic epithelial tumors, we identified 4208 mutations. We constructed a concordance table and observed that the GTF2I and TP53 mutations are mutually exclusive. While GTF2I mutations were common in A and AB thymomas, TP53 mutations were mainly observed in B3 thymomas and thymic carcinomas. Mutations in GTF2I, HRAS, TTN, and UNC93B1 tended to co-occur in the same tumors. On the other hand, CYLD, KIT, BRCA2, SETD2, PBRM1, and CDKN2A mutations significantly co-occurred with those of TP53 [13]. As expected, TP53 mutations are associated with poor survival in thymic epithelial tumors [8].

Sequencing of metastatic, pretreated thymic carcinomas revealed mutations in epigenetic regulatory genes, a typical feature of advanced cancers [8]. In thymic carcinomas, mutations were found in genes involved in the histone modification pathway, including BAP1 (6%), SETD2 (11%), ASXL1 (4%), and in chromatin remodeling genes, such as SMARCA4 (4%). Additionally, mutations in genes associated with DNA methylation were observed, including DNMT3A (7%), TET2 (4%), and WT1 (4%) [8].

Using CGH, Zettl and colleagues demonstrated that different histotypes of thymic epithelial tumors exhibit distinct patterns of copy number aberrations [108]. A-type thymomas exhibit only occasional aberrations, whereas B3 thymomas and thymic carcinomas display frequent arm-level copy number losses involving chromosomes 6, 3p, and 13q, as well as copy number gains affecting chromosomes 1q, 7, and 20p [12]. The most common abnormality in thymic carcinomas is the copy number loss of chromosome 16q [10]. Gene fusions resulting from translocations or intrachromosomal rearrangements have been reported in thymic epithelial tumors [10,12,109]. The BRD4-NUT translocation t(15;19) is characteristic of NUT carcinoma of the thorax and identifies a subset of highly aggressive thymic carcinomas [7]. Subsequently, the MCM4-SNTB gene fusion was described in a thymic adenocarcinoma [110]. Gene fusion of KMT2A and MAML2 has been described in 11 thymomas out of 242 thymic epithelial tumors. All these thymomas were B2 or B3, with only one case showing foci of thymic carcinoma [111]. Recurrent fusion genes were not observed in RNA sequencing analyses of thymic carcinomas [12].

### 4.9. Avoid Immune Destruction

The immunosurveillance theory suggests that the immune system recognizes and destroys cancer cells, preventing the growth of clinically detectable tumors. T-lymphocytes and natural killer cells provide this protection, as demonstrated by the adoptive tumor destruction observed after transplantation of cellular-mediated immunity in mice [76]. The predisposition of immunocompromised patients to develop neoplasms and the effectiveness of anti-neoplastic therapies with immune checkpoint inhibitors support this theory. Hence, a neoplasm must have the ability to evade the control of the immune system in order to grow. However, thymic epithelial tumors originate from cells that typically interact with lymphocyte precursors, guiding them through positive selection to recognize antigens and helping them avoid the recognition of self-antigens during negative selection. Therefore, some thymic epithelial cells must be able to prevent the activation of thymocytes when they recognize antigens presented by the MHC through their TCR. Programmed death-1 (PD-1) inhibits TCR-mediated positive selection via interactions with its ligand, PD-L1 [112]. Thymocytes express PD1 during multiple stages of their maturation. Thymic epithelial cells express PD-L1 and PD-L2. PD-L2 is restricted to the medulla, whereas PD-L1 is expressed throughout the thymus [113]. In certain cases, the recognition of self-antigens can lead to the polarization of T lymphocytes towards a regulatory T cell (Treg) phenotype. Assessing lymphocyte infiltration into thymic epithelial tumors, including Tregs, and evaluating the expression of co-stimulatory molecules can be challenging due to the normal physiological function of the thymus. Nevertheless, tumors with GTF2I mutations exhibited lower immune cell infiltration, including M2 macrophages, activated mast cells, neutrophils, plasma cells, T helper follicular cells, and activated memory CD4 T cells. Furthermore, PD-1, PD-L1, and CTLA4 expression were lower in GTF2I mutants than in wild-type tumors [114]. In non-small-cell lung cancers, the expression of PD-L1 is a predictive factor for response to immunotherapy. The expression of PD-L1 has been assessed through immunohistochemistry in thymic epithelial tumors, but inconsistent results have been reported by various authors. The use of several different PD-L1 immunohistochemical tests employing various antibodies, different definitions of PD-L1 positivity, and cutoff values may explain the inconclusive results. The expression of PD-L1 ranges from 23% to 92% in thymoma and 36% to 100% in thymic carcinoma [78,115,116,117,118,119,120,121,122,123,124,125,126,127,128,129,130,131,132]. Unsurprisingly, anti-PD-1 or PD-L1 therapy in thymomas can lead to severe immune-related adverse events [133]. Thymic carcinomas exhibit a higher number of mutations compared to thymomas, and treatment with pembrolizumab has demonstrated objective responses in 23% of cases, including some long-lasting responses. Even in thymic carcinomas, some patients may experience specific immune-related adverse events, such as myocarditis, and thus require close monitoring [130]. The presence of BAP1 mutations serves as a predictive factor for resistance to immunotherapy in thymic carcinomas. BAP1 mutations are associated with low PD-L1 expression in thymic carcinoma. Conversely, CYLD mutations were predictive factors for a response to pembrolizumab. CYLD mutations are linked to high PD-L1 expression in thymic carcinoma, and the downregulation of CYLD is associated with PD-L1 expression mediated by interferon-gamma in thymic epithelial tumor cell lines [134].

Among other co-stimulatory molecules that can interact with the activation of the T cell response, ICOS and CTLA-4 were commonly expressed in thymic epithelial tumors, with a 91% expression rate for each. Furthermore, a higher expression was observed in 48% of ICOS and 52% of CTLA-4 [116]. A higher expression of CTLA-4 is a poor prognostic factor in thymomas and identifies a subset of patients with a lower incidence of myasthenia gravis [135]. In thymic carcinomas, the expression of the Treg markers IDO and FOXP3 was detected in 14% and 29% of patients, respectively. Lower IDO expression and high FOXP3 Treg expression were independent favorable prognostic factors [130]. In solid tumors, CD70 is exclusively expressed on the tumor cells, facilitating immune evasion through interaction with CD27 expressed in the microenvironmental cells [136]. The majority of thymic carcinomas (87%) express CD70, whereas all thymic carcinoids and thymomas are CD70-negative [137].

### 4.10. Tumor-Promoting Inflammation

Necrosis is a common feature of thymic carcinomas, but it is rarely observed in thymomas [138]. Necrosis leads to inflammation and modifies the tumor microenvironment through the production of chemokines. 

During inflammation, macrophages, when polarized to the M2 phenotype, produce growth factors that promote and sustain tumor proliferation and hinder T-cell-mediated destruction of the tumor [76]. A more prominent infiltration of M2 macrophages (CD204+) has been associated with a poorer prognosis for thymic epithelial tumors [138]. Moreover, cancer cells can directly enhance inflammation through the release of specific factors. HMGB1 is a non-histone chromosomal protein that functions as a DNA chaperone. When HMGB1 is released from the cell, it functions as a damage-associated molecular pattern molecule (DAMP). The activation of receptors for advanced glycation end products (RAGE) enhances chronic inflammation, a pathologic condition that promotes the growth of epithelial malignancies [139]. Moreover, RAGE increases the resistance of tumor cells to hypoxia and promotes tumor invasiveness and metastatic spread [140]. Several ligands, including HMGB1, can activate RAGE. Thymic epithelial tumors commonly express RAGE in their cytoplasm, with higher expression observed in thymic carcinomas and B2 thymomas. On the contrary, thymic carcinomas express less HMGB1 compared to thymomas [141]. HMGB1 regulates apoptosis and autophagy at multiple cellular levels [142,143]. Extracellular HMGB1 promotes autophagy by activating RAGE, whereas cytoplasmic HMGB1 binds to Beclin1. Nuclear HMGB1 regulates autophagy through heat shock protein 27 (HSP27/HSPB1) [142]. Conversely, heat shock protein 70 (HSP70) activates NFκB and JNK/Beclin-1, establishing a positive feedback mechanism that leads to increased expression and release of HMGB1 [144]. In thymic tumors, HSP70 and HSP27 were expressed in both epithelial and microenvironmental cells [145].

## 5. Conclusions

Thymic carcinomas are rare but display heterogeneous histological characteristics. The majority are composed of squamous cell carcinomas. Thymic carcinomas differentiate from thymomas in terms of histological appearance and prognosis, as they are tumors that tend to be notably aggressive. At a molecular level, thymomas and thymic carcinomas appear as two distinct entities. Thymomas frequently exhibit mutations in GTF2I, while thymic carcinomas have mutations in TP53, CDKN2A/B, and, in advanced forms, mutations in genes that control chromatin modifications and epigenetic regulation processes. Immunotherapy and drugs with anti-angiogenic properties have demonstrated the significance of these two hallmarks of cancer in the biology of these tumors. Other molecular aspects that support the growth of thymic carcinomas require further analysis in order to develop even more effective therapies.

## 6. Future Directions

A more profound comprehension of the hallmarks of cancer in thymic carcinoma is necessary for the identification of targets for effective therapeutic interventions. Novel hallmarks are emerging and are under evaluation for formal proposals, including epigenetic regulation and neuronal infiltration of tumors, among others. Oncological treatments are evolving rapidly with the discovery of new targets and the development of more efficient drugs. For instance, drug-conjugated antibodies will offer new treatment opportunities for patients who are not suitable candidates for local therapies.

## Figures and Tables

**Table 1 cancers-16-00166-t001:** Classification of thymic carcinomas.

Thymic Carcinomas	Proportion	Peculiar Molecular Features
Squamous cell carcinomas (SSC) *Subtype SSC: Micronodular thymic carcinoma with lymphoid hyperplasia*	70–80%	
Basaloid carcinoma of the thymus	<5%	
Lymphoepithelial carcinoma of the thymus	1.3–6%	EBV infection in half of patients
NUT carcinoma of the thorax	<1% (about 200 cases)	t(15:19) NUT-BRD4 fusion gene (NUT can rearrange with other partners)
Clear cell carcinoma of the thymus (CCC) *Subtype CCC: Hyalinizing clear cell carcinoma*	Very rare (about 25 cases)	EWSR1-ATF1 gene fusion characterizes salivary gland-type hyalinizing clear cell carcinoma
Low-grade papillary adenocarcinoma of the thymus	3%	
Mucoepidermoid carcinoma of the thymus	2.50%	Occasionally associated with CRTC1-MAML2 gene fusion
Thymic carcinoma with adenoid cystic carcinoma-like features	Very rare (<10 cases)	
Enteric-type adenocarcinoma of the thymus	<5%	
Adenocarcinoma NOS of the thymus	1.6% (about 70 cases)	
Adenosquamous carcinoma of the thymus	unknown	
Sarcomatoid carcinoma of the thymus	2.5–10%	
Undifferentiated carcinoma of the thymus	2.50%	
Thymic carcinoma (NOS)		

**Table 2 cancers-16-00166-t002:** KIT mutations described in thymic carcinomas.

KIT Mutations in TC	References	Exon	Therapy, Objective Response
E490K	[39]	exon 9	Imatinib
M552Nfs*13	[8]	exon 10	
Y553N	[40,41]	exon 11	Imatinib, PR; sunitinib PR
T574del	[28]	exon 11	
Q575*	[28]	exon 11	
W557R	[39]	exon 11	Imatinib
V559A	[39]	exon 11	Imatinib
V559A	[42]	exon 11	
V559G	[42]	exon 11	
V560del	[43]	exon 11	Imatinib
V560del	[44]	exon 11	Imatinib, SD
E561K	[28]	exon 11	
L576P	[39]	exon 11	Imatinib
L576P	[45]	exon 11	
L576P	[42]	exon 11	
L576P	[46]	exon 11	
L576P	[8]	exon 11	
L576P	[46]	exon 11	
P577_Y579del	[47]	exon 11	Sorafenib, PR
D579del	[48]	exon 11	Imatinib, SD
D579del	[8]	exon 11	
R586K	[28]	exon 11	
R588M	[13]	exon 11	
G601W	[8]	exon 12	
H697Y	[43]	exon 14	Sorafenib
D820E	[49]	exon 17	Sorafenib, PR
D820E	[42]	exon 17	
N822K	[50]	exon 17	Avapritinib, SD
Y823C	[50]	exon 17	Avapritinib, PD
Y823D	[9]	exon 17	(B3 Thymoma)
Y823S	[46]	exon 17	
Y823S	[46]	exon 17	

* indicates a stop codon.

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
