# Peer review of "Molecular and Functional Key Features and Oncogenic Drivers in Thymic Carcinomas"

_cancers, 2023, doi:10.3390/cancers16010166_

Round 1

Reviewer 1 Report

Comments and Suggestions for Authors

Dear Authors,

well elaborated and informative manuscript with very large literature sources. I found it very useful and have only some minor comments:

1) Introduction part is good, but please find please a place and insert a paragraph with the aim and plan of the review;

2) Material and method section is not a mandatory, but I would like to ask you to enroll also the period of work, data bases used, key words used, literature sources inclusion/exclusion criteria. This always is of help to understand the following information better!

3) the text and subsections are developed in a logic way and gives an info for the reader, also conclusions and future directions are written in good manner. Just a little remark again with the References, - could you, please, remove or / and replace those 8 from the previous century? Not all of them seemed so important for your manuscript...

Otherwise thank you for the logic and informative manuscript, after the corrections I definitely advice to publish this work!

Author Response

We apricate your effort to improve our article with your comments. Please find below point to point answers.

Question 1) Introduction part is good, but please find a place and insert a paragraph with the aim and plan of the review.

Answer 1) To better clarify the aim of the review we included the following statement at the end of the introduction section: Given the rarity of the disease and the heterogeneous and non-systematic scientific production in recent years, we aim to examine the recent literature on molecular alterations in thymic carcinomas. This review will be organized according to the well-established hallmarks of cancer in order to understand which of these have been well-characterized and which still represent perspectives for innovative research.

Question 2) Material and method section is not a mandatory, but I would like to ask you to enroll also the period of work, data bases used, key words used, literature sources inclusion/exclusion criteria. This always is of help to understand the following information better!

Answer 2) We included in the review a material and methods section: The literature review was conducted using the PubMed database. The search was performed with the keywords 'thymic carcinoma' and 'thymic epithelial tumors.' Articles published between 2023 and 2000 were included if they reported molecular aberrations in thymic carcinomas. We specifically focused on references related to the hallmarks of cancer, as described in 2011. Epigenetic abnormalities, including micro-RNA, long noncoding RNA, and methylation, were excluded unless directly associated with a specific hallmark of cancer. Additionally, data included in the fifth edition of the World Health Classification of Tumors: Thoracic Tumors were incorporated. The data predominantly pertain to squamous cell carcinomas of the thymus; specific aberrations of other types of thymic carcinomas are poorly understood and are excluded from this review.

Question 3) The text and subsections are developed in a logic way and gives an info for the reader, also conclusions and future directions are written in good manner. Just a little remark again with the References, - could you, please, remove or / and replace those 8 from the previous century? Not all of them seemed so important for your manuscript...

Answer 3) We removed the references published before the 2000.

Reviewer 2 Report

Comments and Suggestions for Authors

This review article entitled "Molecular and functional key features and oncogenic drivers in thymic carcinomas (cancers-2734283)" by Dr. Serena Barachini reviewed the clinicogenomic entity as for thymic malignancies related to chemotherapy and immunological complication including myasthenia gravis. This review included genetic alteration, transcriptome, metabolism, cancer immunity in landscape of cancer section in thymoma and thymic carcinoma. This review is updated and comprehensive, therefore this review will be helpful for readers. The lack of this review is lack of view of treatment, but I think there is not enough space to cover this topic. Also, this review covers updated references.

Minor comments

Page 2, line 46: Add the following information: lymphoepithelioma-like carcinoma is associated with EBV. Also, change the current version to lymphoepithelioma-like carcinoma. This is already described in Table 1.

On page 2. Please clarify the frequency of TMB in thymoma and thymic carcinoma.

Author Response

We apricate your effort to improve our article with your comments. Please find below point to point answers.

Question 4) Page 2, line 46: Add the following information: lymphoepithelioma-like carcinoma is associated with EBV. Also, change the current version to lymphoepithelioma-like carcinoma. This is already described in Table 1.

Answer 4) We included in the text the following sentence: Lymphoepithelioma-like carcinoma is associated with EBV infection similarly to what observed in nasopharynx tumors.

Question 5) On page 2. Please clarify the frequency of TMB in thymoma and thymic carcinoma.

Answer 5) Thank you for pointing out this problem. We removed the frequency of mutations because inconsistent between different reports. The values previously indicated 3.84 and 1.92 mut/Mb (according to the article published by Girard et al.) overestimated results of exome sequencing provided by TCGA. However, in all reports TMB is higher in thymic carcinoma than in thymomas.

Reviewer 3 Report

Comments and Suggestions for Authors

This paper is well written and the content is also very rich.

Comments: 

1. Although the most common histological type of thymic cancer is squamous cell carcinoma, there are also other rare types, and molecular biological characteristics may vary, So if this article only discusses thymic squamous cell carcinoma, it may be more focused.

2. Most tumor targeting and immunotherapy are for advanced patients, and this article did not discuss according to the different staging disease. Is there a difference in molecular biological markers among different disease stages?

Author Response

We apricate your effort to improve our article with your comments. Please find below point to point answers.

Question 6) Although the most common histological type of thymic cancer is squamous cell carcinoma, there are also other rare types, and molecular biological characteristics may vary, So if this article only discusses thymic squamous cell carcinoma, it may be more focused.

Answer 6) We introduced in the text a materials and methods section and we explained that the results were mainly related to squamous cell carcinomas. The other types of carcinomas are only described in the introduction and listed in table 1 for knowledge of the reads.

Question 7) Most tumor targeting, and immunotherapy are for advanced patients, and this article did not discuss according to the different staging disease. Is there a difference in molecular biological markers among different disease stages?

Answer 7) Thymic carcinomas are commonly diagnosed in advanced stages III-IV. Even if rare we have several data regarding of the molecular diagnostic of resected (stage I-II) thymic carcinomas from TCGA study. Comparing data from advanced tumors and from earlier stages we can observe a higher incidence of mutations in genes involved in the epigenetic regulation. We stated this observation in chapter regarding the instability of genome of thymic carcer cells: Sequencing of metastatic, pretreated thymic carcinomas revealed mutations in epigenetic regulatory genes, a typical feature of advanced cancers. In thymic carcinomas, mutations were found in genes involved in the histone modification pathway, including BAP1 (6%), SETD2 (11%), ASXL1 (4%), and in chromatin remodeling genes, such as SMARCA4 (4%). Additionally, mutations in genes associated with DNA methylation were observed, including DNMT3A (7%), TET2 (4%), and WT1 (4%).